# Acute stress symptoms 1–2 weeks after stroke predict the subsequent development of post-traumatic stress symptoms: A prospective cohort study

**David Feely[1], Brian Slattery[1], Thomas Walsh[2], Trish Galvin[2], Kate Donlon[2], Michelle Hanlon[1], Darina Gormley[1], Gwen-Marie Brown[1], Sarah Quinn[1], Stephanie Robinson[2], Conor Judge[2], Martin O'Donnell[2], Kiran Sarma[1], Brian E. McGuire[1,2]***

**1** School of Psychology, University of Galway, Galway, Ireland, **2** Department of Stroke and Geriatric Medicine, University Hospital, Galway, Ireland

* brian.mcguire@nuigalway.ie

**Data Availability Statement:** All relevant data are within the manuscript and its Supporting information files.

## Abstract

### Objective

To date no research has examined the potential influence of acute stress symptoms (ASD) on subsequent development of post-traumatic stress disorder (PTSD) symptoms in stroke survivors. Our objective was to examine whether acute stress symptoms measured 1–2 weeks post-stroke predicted the presence of post-traumatic stress symptoms measured 6–12 weeks later.

### Design

Prospective within-groups study.

### Methods

Fifty four participants who completed a measure of acute stress disorder at 1–2 weeks following stroke (time 1) and 31 of these participants completed a measure of posttraumatic stress disorder 6–12 weeks later (time 2). Participants also completed measures of stroke severity, functional impairment, cognitive impairment, depression, anxiety, pre-morbid intelligence and pain across both time points.

### Results

Some 22% met the criteria for ASD at baseline and of those, 62.5% went on to meet the criteria for PTSD at follow-up. Meanwhile two of the seven participants (28.6%) who met the criteria for PTSD at Time 2, did not meet the ASD criteria at Time 1 (so that PTSD developed subsequently). A hierarchical multiple regression analysis indicated that the presence of acute stress symptoms at baseline was predictive of post-traumatic stress symptoms at

**Funding:** The author(s) received no specific
funding for this work.

**Competing interests:** The authors have declared
that no competing interests exist.

follow-up ($R^2$ = .26, p < .01). Less severe stroke was correlated with higher levels of post-traumatic stress symptoms at Time 2 (rho = .42, p < .01).

## Conclusions

The results highlight the importance of early assessment and identification of acute stress symptoms in stroke survivors as a risk factor for subsequent PTSD. Both ASD and PTSD were prevalent and the presence of both disorders should be assessed.

## Introduction

The Diagnostic and Statistical Manual of Mental Disorders, 5$^{th}$ Edition [1] recognizes the potential to develop trauma and stressor-related disorders following exposure to events where risk of death or serious injury is experienced and this includes sudden, catastrophic medical events. Two of the key disorders in this category are Acute Stress Disorder (ASD) and Post-Traumatic Stress Disorder (PTSD). These two disorders are characterized by four groups of symptoms (a) intrusive memories, dreams, flashbacks, dissociative symptoms (b) avoidance of situations, memories or symbols of the traumatic event (c) hyperarousal, anxiety and overreactivity including sleep disturbance and irritable or angry outbursts (d) negative cognitions and negative mood. However, recent reviews highlighted that relatively little is known about the development of specific types of anxiety [2] and traumatic stress reactions following stroke [3].

ASD is differentiated from PTSD in the psychiatric classification system primarily by virtue of the duration of the symptoms. The symptoms of ASD must persist for at least 3 days and must resolve within one month of the trauma—if the symptoms persist for more than one month, then PTSD may be diagnosed. It is possible that PTSD may develop at any stage after a traumatic event, although normally symptoms appear within a few months of the event [1]. It is not necessary for symptoms to have been present immediately after the event—in other words, it is not necessary for ASD to have been present in order for PTSD to develop subsequently.

Previous studies suggest that one in four people may develop PTSD after a stroke [4] and factors such as younger age [5] and co-morbid depression and anxiety [6] are associated with higher risk of PTSD post-stroke. As psychological difficulties can impact on functional outcomes [7] and quality of life in stroke survivors [8, 9], a call has been made for longitudinal studies to aid in the early identification of risk factors for PTSD in the aftermath of stroke [10].

One potential risk factor for the development of PTSD is the presence of ASD symptoms [11], but it is not known if this is relevant for risk of PTSD after stroke. Thus far there is an absence of research examining (a) the extent to which ASD manifests after a stroke and (b) the potential influence of ASD symptoms as a risk factor for developing PTSD in stroke survivors. Psychological intervention to identify and treat acute stress symptoms after stroke may reduce the potential for these symptoms to develop into a more chronic psychological condition.

### Aims and hypotheses

Our aims were: (a) examine the prevalence of ASD and PTSD following stroke (b) examine whether demographic (i.e., age, gender, nationality, marital status, living situation and years of education) or clinical variables (i.e., ASD diagnosis, functional impairment, cognitive impairment, depression, anxiety, pain, premorbid intelligence, stroke severity, stroke type, previous stroke, or history of mental illness) predict the development of PTSD. In particular, it

was hypothesized that higher levels of ASD symptoms at 1–2 weeks post stroke would predict higher levels of PTSD symptoms at 6–12 weeks post-stroke.

## Method

### Design

A prospective within-groups design to investigate whether acute stress symptoms measured at baseline predicted post-traumatic stress symptoms at follow-up. Time 1 (baseline) data collection took place within two weeks of a patient having a stroke (diagnostic criteria for ASD require symptoms to have developed within one month of trauma). Time 2 data collection took place between 6–12 weeks post stroke (diagnostic criteria for PTSD require that symptoms have persisted for more than one month).

Patients were enroled over a 12-month period at the Stroke Unit of a tertiary hospital which serves a mixed urban and rural catchment area of approximately 1 million people. Inclusion criteria were (1) over 18 years (2) competent in English (3) confirmed diagnosis of stroke by a stroke physician (4) able to provide informed consent to participate across two time points. Exclusion criteria were (1) too medically unwell as determined by a stroke physician (2) severe cognitive or sensory impairment that would prevent completion of cognitive testing (3) severe communication impairment.

From study inception date, 212 patients admitted were assessed as having a confirmed diagnosis of stroke. Of these, 107 (50.5%) patients were assessed as not being suitable to participate in the study: 7 patients (6.6%) were not adequately fluent in English, 28 (26.2%) had severe cognitive difficulties, 5 (4.7%) had significant hearing or visual difficulties, 23 (21.5%) had aphasia, and 44 (41.12%) were too medically unwell to participate.

Of the remaining 105 patients who were deemed suitable for participation, 54 (51.4%) were assessed at Time 1. Meanwhile, 26 patients (24.8%) declined participation and 25 (23.8%) participants were discharged from the stroke unit prior to being approached by the research team. These patients were either discharged home within 24 hours or to another hospital, which was not accessible by the research team. It was possible to follow up with 31 of the initial 54 patients (57.4%) at 6–12 weeks. Of the 23 participants for whom it was not possible to collect Time 2 data, 7 declined participation, 11 did not respond to the invitation, 4 were too unwell, and 1 was deceased.

The remaining sample of 54 participants included 36 (66.7%) males and 18 females with an average age for the entire sample of 66.15 years (SD = 6.32), (range 34–96 years, median 70) (see summary Table 1). Most of the participants were Irish (87%), the remainder were from other EU countries and 74% lived with another person.

### Primary outcome measures

**The acute stress disorder scale.** The Acute Stress Disorder Scale (ASDS) for DSM-IV [12] was used to measure symptoms of acute stress. The ASDS is a 19-item measure of acute stress experiences following traumatic events based on DSM-IV criteria [13]. While the most recent version of the DSM is the 5[th] edition [1], we utilized a measure of ASD based on DSM-4 because we could not identify a validated patient-reported measure of ASD based on DSM-5 criteria. In order for a diagnosis of ASD to be made, a person must have experienced a stressor and respond with fear and helplessness (Criterion A), have at least three dissociative symptoms (Criterion B), at least one re-experiencing symptom (Criterion C), marked avoidance (Criterion D) and anxious arousal (Criterion E). A symptom is considered present if an individual scores 3 or more on the item that corresponds to that symptom. The ASDS demonstrates good

**Table 1. Demographic characteristics of the sample.**

|  | No. of Participants (n = 54) | % |
|---|---|---|
| **Gender** |  |  |
| Male | 36 | 66.7 |
| Female | 18 | 33.3 |
| **Nationality** |  |  |
| Irish | 47 | 87 |
| Other EU Country | 7 | 13 |
| **Marital Status** |  |  |
| Single | 8 | 14.8 |
| In a Relationship/ Married | 31 | 57.4 |
| Separated/ Divorced | 7 | 13 |
| **Living Situation** |  |  |
| With someone | 40 | 74.1 |
| Alone | 14 | 25.9 |
| **Stroke Type** |  |  |
| Ischaemic | 46 | 85.2 |
| Haemorrhagic | 4 | 7.4 |
| TIA | 4 | 7.4 |
| **Previous Stroke** |  |  |
| Yes | 11 | 20.4 |
| No | 43 | 79.6 |
| **History of Mental Ill-Health** |  |  |
| Yes | 10 | 18.5 |
| No | 44 | 81.5 |
| **Education Level** |  |  |
| None | 1 | 1.9 |
| Primary School | 19 | 35.2 |
| Secondary School | 25 | 46.3 |
| Third Level | 9 | 16.7 |

sensitivity (95%) and specificity (83%) and has a robust test-retest reliability score (r = .94) [12]. Cronbach's alpha for the ASDS in the current study was α = .85.

**The PTSD checklist.** The PTSD Checklist (PCL-S) [14] measures the 17 symptoms of PTSD as defined by DSMIV. We used the DSM-4 version to align with the version of ASDS used above. A score of 3 or more indicated the presence of a symptom and we used the recommended cut-off score of 36–44 [15]. The PCL-S has good sensitivity (94%) and specificity (86%) [16]. Cronbach's alpha for the PCL-S in the current study was α = .90.

## Secondary outcome measures

**The hospital anxiety and depression scale.** The Hospital Anxiety and Depression Scale (HADS) [17] was used to measure anxiety and depression. Each of the 14 items is scored 0–3 with higher scores indicating greater symptoms of anxiety and depression. The HADS is validated for use with stroke patients [18]. In the current study, Cronbach's alpha for the HADS-A was α = .72 and α = .60 for the HADS-D.

**Repeatable battery of neuropsychological tests.** The Repeatable Battery of Neuropsychological Tests (RBANS) [19] is a neuropsychological test battery that measures performance

across five domains: immediate memory, visuospatial/constructional ability, language, attention and delayed memory. We used a cutoff score of <83 on the total scale as recommended for stroke patients [20].

**The Montreal Cognitive Assessment.** The *Montreal Cognitive Assessment* (MoCA) [21] was used in addition to the RBANS to measure cognitive impairment. Scores >26 are considered normal, with scores falling below this indicating cognitive impairment. The MoCA has been shown to be a valid measure of post-stroke cognitive impairment [22]. The alpha coefficient for the measure was reported as .83 [22].

**Barthel Activities of Daily Living Index.** The Barthel Activities of Daily Living Index [23] was used to measure impairment in activities of daily living across ten activity areas: feeding, bathing, grooming, dressing, bowel, bladder, toilet use, transfers, mobility, and stairs. The scores are summed to provide a total score (/100) with three categories of functional impairment: severe (0–50), moderate (51–75) and mild (76–100). The Barthel has been widely used with stroke patients [6]. Cronbach's Alpha in the current study was α = .85.

**The Scandinavian Stroke Scale.** The Scandinavian Stroke Scale (SSS) [24] was used to assess stroke severity. This 9 item measure assesses symptom severity across the following areas: somnolence, eye movement, motor power (arm, hand, leg), orientation, speech, facial palsy, gait. These items are summed and higher scores indicate less severity: 0–18 very severe, 19–32 severe, 33–44 moderate, 45–58 mild stroke [25]. Cronbach's alpha for the SSS was α = .65 in the current study.

**The Brief Pain Inventory-S.** The Brief Pain Inventory (BPI-S) [26] was used to measure (a) pain intensity (b) pain interference with higher scores indicating more difficulties. The Cronbach's alphas in the current study were α = .92 and α = .95, for pain intensity and interference, respectively.

**National Adult Reading Test.** The National Adult Reading Test 2nd Edition (NART) [27] was used to estimate pre-morbid intelligence. The NART consists of a list of 50 English words presented in order of increasing difficulty. The participant reads each word aloud and the number of errors are recorded, providing a total error score out of 50. The alpha coefficient for the scale has been reported at .93 [27].

**Procedure.** First, patients were assessed by a Clinical Nurse Specialist or Stroke Registrar as being potentially suitable for inclusion in the study and this was then confirmed by a medical stroke specialist. Suitable participants were then approached by a member of the team in order to outline the nature of the study and invite them to take part. If agreeable, participants signed a consent form and were informed that they were free to withdraw consent at any time. Time 1 assessment then took place at the patient's bedside. The ASDS was administered at Time 1 along with the other measures outlined above. The order of the assessment was the same for each participant.

After 4–6 weeks, participants were reassessed using the same battery as Time 1 and the PCL-S was administered instead of ASDS (stroke severity was not measured again).

**Statistical strategy.** To examine trends in the data, descriptive and correlation statistics were computed. Mann Whitney tests were used to compare those who did/did not meet criteria for ASD and PTSD at Time 1 and Time 2 respectively. A hierarchical linear regression was conducted in order to investigate the hypothesis that higher levels of acute stress symptoms at Time 1 (ASDS) would predict higher levels of post-traumatic stress symptoms at Time 2 (PCL-S). Where data violated assumptions of the relevant tests, then they were either transformed or analysed using non-parametric alternatives. Effect sizes were calculated and converted to Pearson's r for ease of interpretation.

**Ethical considerations.** The study was approved by the Research Ethics Committee of University Hospital Galway, Ireland. The study complied with the provisions for good ethical practice and all participants gave written consent to participate.

## Results

The demographic variables are presented in Table 1.

### Comparison of baseline and follow-up clinical measures

The clinical measures at baseline and follow-up are summarized in Table 2.

The mean time between the initial stroke and baseline assessment was 5.9 days (SD = 3), and 7.4 weeks (SD = 1.5) for the follow-up assessment. Results from the Mann Whitney U test revealed a significant improvement in scores on the Barthel Index from baseline to follow-up ($U = 599$, $z = -2.25$, $p < 0.05$, r = -.24) and a significant improvement on the RBANS at follow-up ($U = 481$, $z = -2.78$, $p < 0.01$, r = -.31) with improvements noted in both immediate and delayed memory. However, pain intensity scores were significantly higher at Time 2 ($U = 672$, $z = -2.16$, $p < 0.05$, r = -.24) (see Table 2).

### Frequency of ASD and PTSD and other clinical cut-off scores

Table 3 summarises the proportions at each time point that were above the clinical cut-off on the measures of ASD, PTSD, stroke severity, depression, anxiety, functional ability, cognitive ability and pain. Of those who completed both baseline and follow-up assessments (n = 31), 8

**Table 2. Comparison of Time 1 and Time 2 participants on each of the clinical measures, along with means, standard deviations, medians, range and proportions with ASD and PTSD.**

| Measure | Mean | Time 1 (n = 54) | | | Mean | Time 2 (n = 31) | | | U | z |
|---|---|---|---|---|---|---|---|---|---|---|
| | | SD | Mdn | Range | | SD | Mdn | Range | | |
| ASDS (ASD) | 35.4 | 12.4 | 19 | 19–70 | - | - | - | - | - | - |
| PCL-S (PTSD) | - | - | - | - | 30.6 | 11.9 | 17 | 74 | - | - |
| HADS Anxiety | 5.7 | 3.9 | 5.5 | 0–14 | 5.5 | 3.6 | 5 | 0–14 | 790 | -.43 |
| HADS Depression | 4.2 | 3.3 | 4 | 0–12 | 4.5 | 3.4 | 4 | 0–13 | 787 | -.46 |
| Barthel Index | 81.7 | 19.7 | 90 | 30–100 | 92.4 | 10.1 | 95 | 65–100 | 599* | -2.25 |
| Stroke Severity Scale (SSS) | 52.8 | 5.5 | 39 | 39–58 | - | - | - | - | - | - |
| Pain Intensity | .6 | 1.8 | 0 | 0–7.5 | 1.6 | 2.5 | 0 | 0–6.5 | 672* | -2.16 |
| Pain Interference | .2 | .8 | 0 | 0–4.2 | .2 | .7 | 0 | 0–3.2 | 805 | -.55 |
| NART | 34 | 11.1 | 36 | 3–50 | 36.8 | 8.6 | 35.5 | 13–50 | 600 | -.66 |
| MOCA | 21.9 | 4.3 | 22 | 11–30 | 23.6 | 3.4 | 24 | 16–30 | 617 | -1.79 |
| RBANS | | | | | | | | | | |
| • Total | 74.5 | 13.3 | 72 | 43–103 | 83.5 | 12.6 | 81.5 | 61–108 | 481** | -2.78 |
| • Immediate Memory | 75.1 | 18.6 | 80 | 40–120 | 88.9 | 18.2 | 90 | 61–123 | 492** | -3.06 |
| • Visuospatial | 83 | 20.5 | 84 | 50–136 | 85.2 | 18.1 | 84 | 53–121 | 695 | -.69 |
| • Language | 85.7 | 12.1 | 88 | 47–112 | 89.3 | 11.7 | 90 | 57–113 | 705 | -1.08 |
| • Attention | 80.6 | 18.9 | 85 | 40–112 | 88 | 16.2 | 86.5 | 60–138 | 658 | -1.05 |
| • Delayed Memory | 75.2 | 18.5 | 78 | 44–110 | 83.7 | 18.1 | 81 | 52–115 | 573* | -1.88 |

Note.

*p < 0.05

**p < 0.01

**Table 3. Percentage frequency distribution of clinical variables.**

|  | Time 1 (n = 54) | Time 2 (n = 31) |
|---|---|---|
| ASD (ASDS; Time 1 Only) | 12 (22.2%) | - |
| PTSD (PCL-S; Time 2 Only) | - | 7 (22.6%) |
| Anxiety (HADS-A) | 18 (33.3%) | 7 (22.6%) |
| Depression (HADS-D) | 9 (16.7%) | 7 (22.6%) |
| Any Pain Reported (BPI-S) | 7 (13%) | 10 (32.3%) |
| Stroke Severity (SSS; Time 1 Only) |  |  |
| • Mild | 49 (90.7%) | - |
| • Moderate | 5 (9.3%) | - |
| Functional Impairment (BI) |  |  |
| • None | 18 (33.3%) | 15 (48.4%) |
| • Mild | 4 (7.4%) | 4 (12.9%) |
| • Moderate | 19 (35.2%) | 12 (38.7%) |
| • Severe | 13 (24.1%) | - |
| Cognitive Impairment (RBANS <83) |  |  |
| • Not Impaired | 10 (18.5%) | 12 (38.7%) |
| • Impaired | 41 (75.9%) | 18 (58.1%) |
| • Unable to complete assessment | 3 (5.6%) | 1 (3.2%) |
| Cognitive Impairment (MoCA <26) |  |  |
| • Not Impaired | 12 (22.2%) | 10 (32.3%) |
| • Impaired | 40 (74.1%) | 21 (67.7%) |
| • Unable to complete assessment | 2 (3.7%) | - |

participants (26%) met the criteria for ASD at baseline and 7 met the criteria for PTSD at fol-low-up (22.6%). Of the specific 8 individuals who met the ASD criteria at baseline, 5 went on to meet the criteria for PTSD and 3 (37.5%) did not—indicating a 62% "conversion rate" from ASD to PTSD. Meanwhile two of the seven participants (28.6%) who met the criteria for PTSD at follow-up, did not meet the ASD criteria at baseline (see Table 3)—therefore around one third became new cases of PTSD with no ASD at baseline.

## Comparison on each of the clinical measures for those who met ASD criteria at baseline

Results from the Mann Whitney U test revealed that those who met ASD criteria at Time 1 had significantly higher depression ($U = 141$, $z = -2.33$, $p < 0.05$, r = -.32) and anxiety scores ($U = 143.5$, $z = -2.27$, $p < 0.01$, r = -.31) than those who did not have ASD. At follow-up, they also had significantly lower overall RBANS scores ($U = 106.5$, $z = -2.60$, $p < 0.01$, r = -.35), indi-cating greater levels of cognitive impairment—specifically in immediate memory ($U = 101.5$, $z = -2.85$, $p < 0.01$, r = -.39) and delayed memory ($U = 105.5$, $z = -2.63$, $p < 0.01$, r = .36).

## Comparison on each of the clinical measures for those who met PTSD criteria at follow-up

We compared those with and without PTSD at follow-up: results from the Mann Whitney U test revealed that those who met PTSD criteria at follow-up also had significantly higher depression ($U = 17.5$, $z = -3.18$, $p < 0.01$, r = -.57) and anxiety ($U = 37.00$, $z = -2.23$, $p < 0.05$, r = -.40) scores at follow-up. They also had significantly lower scores on the RBANS measure of delayed memory ($U = 38$, $z = -.21$, $p < 0.05$, r = -.38).

**Table 4. Linear regression predicting post-traumatic stress symptoms.**

| Block | | Unstandardized Coefficients | | Standardized Coefficients | t | Sig. | $R^2$ | $R^2$ Change |
|---|---|---|---|---|---|---|---|---|
| | | B | Std. Error | Beta | | | | |
| 1 | (Constant) | 49.30 | 13.01 | | 3.79 | 0.001 | | |
| | Gender | -1.75 | 4.53 | -0.07 | -0.39 | 0.70 | | |
| | Age | -0.25 | 0.17 | -0.26 | -1.43 | 0.16 | 0.08 | 0.08 |
| 2 | (Constant) | 37.37 | 16.47 | | 2.27 | 0.03 | | |
| | Gender | 2.00 | 5.53 | 0.08 | 0.36 | 0.72 | | |
| | Age | -0.14 | 0.19 | -0.15 | -0.75 | 0.46 | | |
| | Stroke Severity | 0.60 | 0.51 | 0.28 | 1.17 | 0.25 | 0.12 | 0.05 |
| 3 | (Constant) | 25.73 | 14.49 | | 1.78 | 0.09 | | |
| | Gender | 3.20 | 4.73 | 0.13 | 0.68 | 0.51 | | |
| | Age | 0.00 | 0.17 | 0.00 | 0.03 | 0.98 | | |
| | Stroke Severity | 1.03 | 0.46 | 0.49 | 2.27 | 0.03 | | |
| | Acute Stress Symptoms | 0.48 | 0.15 | 0.54 | 3.33 | 0.001 | 0.38 | 0.26** |
| 4 | (Constant) | 23.67 | 14.11 | | 1.68 | 0.11 | | |
| | Gender | 4.01 | 4.62 | 0.16 | 0.87 | 0.39 | | |
| | Age | 0.03 | 0.17 | 0.03 | 0.16 | 0.87 | | |
| | Stroke Severity | 1.09 | 0.44 | 0.52 | 2.46 | 0.02 | | |
| | Acute Stroke Symptoms | 0.62 | 0.16 | 0.70 | 3.77 | 0.001 | | |
| | Interaction Term | 0.04 | 0.02 | 0.29 | 1.62 | 0.12 | 0.44 | 0.06 |

*$p < .05$,

**$p < .01$

## Bivariate correlations

A correlation analysis was conducted using Spearman's Rho, to analyse the relationships amongst the baseline study variables and the follow-up PCL-S score (n = 31). Higher levels of acute stress symptoms were correlated with higher levels of post-traumatic stress symptoms (rho = .48, p < .01). Higher levels of acute stress symptoms were also correlated with higher levels of anxiety (rho = .67, $p < .01$) and depression (rho = .54, $p < .01$), and with lower RBANS delayed memory scores (rho = -.54, p =.<01). Meanwhile, higher levels of post-traumatic stress was correlated with higher scores on the SSS (i.e. less severe stroke) (rho = .42, p = .01).

## Hypothesis testing

Using hierarchical multiple regression, we examined whether acute stress symptoms at baseline predicted higher levels of post-traumatic stress symptoms at follow-up and the predictive value of the interaction of stroke severity and acute stress symptoms. The two variables were mean-centered to allow both main effects and interaction effects to be included in the model (i.e. to minimise multicollinearity).

The model was tested in four blocks (see Table 4). In Block 1, demographic variables (age, gender) were entered and neither the block as a whole nor the individual variables were significant predictors of post-traumatic stress symptoms. In Block 2, the stroke severity score was not a significant predictor of post-traumatic stress symptoms. The acute stress symptoms score was entered in Block 3 and made a statistically significant contribution to the model over Block 2 ($R^2$ change = .26, p < .01), and when combined with Block 1 and 2 the three Blocks

accounted for 38% variation in post-traumatic stress symptoms, $F(4,26) = 4.02$, p < .05. Acute stress symptoms were a significant predictor of post-traumatic stress symptoms (β(standardized) = .54, p < .01). Stroke severity was also a significant predictor of posttraumatic stress symptoms in Block 3 (β standardized) = .49, p < .05). Finally, the interaction between acute stress symptoms and stroke severity was explored (mean centred ASDS multiplied by mean centred SSS). Results indicated that the addition of the interaction did not contribute to the model.

## Discussion

This study was the first prospective examination of the relationship between acute stress symptoms and post-traumatic stress symptoms in stroke survivors. We assessed whether acute stress symptoms measured 1–2 weeks post-stroke, predicted post-traumatic stress symptoms at 6–12 weeks follow-up. In terms of generalisability, our findings are broadly consistent with other research [28, 29] in which 26% of participants met diagnostic criteria for ASD at baseline. Of those with ASD at time 1, around 65% went on to have PTSD while some 28% developed PTSD at follow-up in the absence of ASD symptoms at baseline. The severity of acute stress symptoms at baseline was associated with greater post-traumatic stress symptoms at follow-up.

These findings have both theoretical and clinical importance. Research with other trauma populations has found that peri-traumatic reactions and dissociation are associated with subsequent PTSD symptoms [30]. Participants who met criteria for both ASD and PTSD had lower overall scores on measures of immediate and delayed memory. Therefore, this study may support the idea that trauma memories are processed in a different way to other memories, consistent with the dual representation hypothesis of PTSD [31].

The finding that less severe strokes were more likely to be associated with PTSD is consistent with the findings of a study that PTSD occurs in around 30% of patients after TIA and is associated with increased risk for other mental health problems [32]. The negative correlation between stroke severity and subsequent PTSD may indicate that the traumatic event is uniquely processed in memory since individuals who are more cognisant throughout the stroke event, may more clearly encode the event in memory. Other research has also shown that conscious memory of the traumatic event (rather than merely being aware that an event happened) increases the likelihood of developing PTSD [33]. However, other research indicates that explicit memory of the traumatic event does not necessarily predict subsequent PTSD symptomatology [34], perhaps supporting the view that traumatic events are processed at an implicit level [35].

The study identified a number of additional challenges amongst this sample of stroke survivors. In line with estimated prevalence rates for anxiety after stroke [2], 33.3% of participants were experiencing anxiety at baseline and 22.6% at follow-up, while 16.7% of participants reported depression at baseline and 22.6% at follow-up. The high rates of comorbidity between PTSD, anxiety and depression [5, 6], might explain why PTSD symptoms have been under-recognized or mis-diagnosed in stroke survivors [36]. The high frequency of these emotional disorders, might also help explain why PTSD is under-recognized in stroke survivors [36]. However, research also suggests that depression and PTSD are independent outcomes of traumatic events that can interact to increase distress and disability [37].

Consistent with previous research, findings indicated that the severity of trauma symptoms did not improve with time [6]. Symptoms of emotional distress can interfere with medication adherence post stroke and have a subsequent effect on rehabilitation [38] and so recognition of symptoms is important.

This study has a number of limitations. First, the sample size was small (especially at Time 2), which reduced the power of the regression analyses. Second, those who were experiencing significant cognitive impairments were not assessed and while this is not uncommon in such studies [2], it may well have impacted on the prevalence of post-traumatic stress symptoms identified. Third, it was not possible to examine personality traits or other facrtors potentially associated with risk for PTSD after stroke. Fourth, we used the DSM-IV criteria for ASD although there is an updated version of DSM available (DSM-V). However, we were unable to identify studies that reported psychometric validation of the DSM-V version of the ASDS. Use of the DSM-IV version may have influenced our findings although a study comparing the capacity of ASD based on DSM-IV and DSM-V to predict PTSD, found that similar proportions of patients were diagnosed with PTSD based on the two definitions of ASD [39]. Furthermore, the diagnoses of ASD and PTSD was confirmed in each case by a clinical psychologist.

While the results suggest that early identification of trauma is important, it may be difficult to routinely identify ASD/PTSD symptoms in busy medical settings and these symptoms are probably under-recognised following stroke [40]. Furthermore, there is evidence that a wide range of anxiety disorders can emerge after stroke including phobias and generalized anxiety [2, 41]. We do not know if general anxiety screening measures are suffficiently sensitive to flag individuals with possible anxiety disorders, so that more detailed assessment can be done. Given the high prevalence of ASD and the consequent risk of developing PTSD, our findings support the use of the ASDS as a potential screen for identifying those at risk of developing PTSD symptoms. Psychological treatments such as CBT for PTSD [42] could then be offered to these patients. The presence of PTSD is itself also a risk factor for subsequent stroke [43] but it is not known whether PTSD that develops after a stroke confers increased risk for another stroke. The results of this study highlight the need for greater awareness of traumatic stress disorders after stroke.

## Supporting information

**S1 File.**
(SAV)

## Acknowledgments

There was no funding for the study. The authors have no conflicts of interest to disclose.

## Author Contributions

**Conceptualization:** David Feely, Brian Slattery, Stephanie Robinson, Brian E. McGuire.

**Data curation:** David Feely, Brian Slattery, Thomas Walsh, Trish Galvin, Kate Donlon, Michelle Hanlon, Darina Gormley, Gwen-Marie Brown, Sarah Quinn, Conor Judge, Brian E. McGuire.

**Formal analysis:** David Feely, Brian Slattery, Kiran Sarma, Brian E. McGuire.

**Investigation:** David Feely, Brian Slattery, Trish Galvin, Kate Donlon, Michelle Hanlon, Darina Gormley, Gwen-Marie Brown, Sarah Quinn, Stephanie Robinson, Conor Judge, Martin O'Donnell, Brian E. McGuire.

**Methodology:** David Feely, Brian Slattery, Michelle Hanlon, Kiran Sarma, Brian E. McGuire.

**Project administration:** David Feely, Brian Slattery, Trish Galvin, Kate Donlon, Michelle Hanlon, Darina Gormley, Gwen-Marie Brown, Sarah Quinn, Brian E. McGuire.

**Resources:** Thomas Walsh, Stephanie Robinson, Conor Judge, Martin O'Donnell.

**Supervision:** Brian Slattery, Brian E. McGuire.

**Writing – original draft:** David Feely, Gwen-Marie Brown, Brian E. McGuire.

**Writing – review & editing:** Brian Slattery, Thomas Walsh, Michelle Hanlon, Gwen-Marie Brown, Kiran Sarma.

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
