## [Decision Letter · Decision Letter 0]

17 Nov 2022

PONE-D-22-27178Acute stress symptoms 1-2 weeks after stroke predict the subsequent development of post-traumatic stress symptoms: A prospective cohort study.PLOS ONE

Dear Dr. McGuire,

Thank you for submitting your valuable work. 

After my own reading, and also based on Reviewer's comments, some concerns need to be clarified and explained. The reviewer's raised essential and relevant comments on the soundness and transparency of the text - which I agree. But, most important, there is a lack of a substantial in-depth debate of the literature. Also, I recommend the authors to carefully read all comments and refine their work both in terms of conciseness and transparency.

Please respond all comments AND highlight them in the text.

We look forward to receiving your revised manuscript.

Kind regards,

Thiago P. Fernandes, PhD

Journal Requirements:

Additional Editor Comments:

Please read my comments.

Reviewers' comments:

Reviewer's Responses to Questions

**Comments to the Author**

1. Is the manuscript technically sound, and do the data support the conclusions?

Reviewer #1: Yes

Reviewer #2: Partly

Reviewer #3: No

2. Has the statistical analysis been performed appropriately and rigorously? 

Reviewer #1: Yes

Reviewer #2: Yes

Reviewer #3: No

3. Have the authors made all data underlying the findings in their manuscript fully available?

Reviewer #1: Yes

Reviewer #2: No

Reviewer #3: No

4. Is the manuscript presented in an intelligible fashion and written in standard English?

Reviewer #1: Yes

Reviewer #2: Yes

Reviewer #3: Yes

5. Review Comments to the Author

Reviewer #1: Appreciate the opportunity to evaluate the manuscript.

I believe that it deals with a very important and is extremely important for scientific innovation in the area.

However, some inconsistencies were presented and I believe they need to be re-evaluated before final approval.

Below I list some of my concerns:

I believe that the abstract is well structured and contains important information for understanding the text. So I have nothing to add on this topic.

In the intro part, some points need to be readjusted. I believe that there was an error in the presentation of the references throughout the text, since they do not follow a citation order, starting in the text with reference four, as presented in the first line of the first paragraph of the introduction.

Also, some moments of the text are missing the citation, I suggest that this be reviewed throughout the manuscript.

Between the second and third paragraph I missed a link link about stroke. Why study this condition? The inclusion of epidemiological data could help in this justification.

Thus, it is important to create a logical line of reasoning during the text. In the third paragraph it starts a new content and in the fourth paragraph it goes back to the content worked in the second. This makes the text confusing. I suggest that authors can readjust, considering the construction of the introduction following the funnel model, starting from the most general subject to the most specific.

I felt that the hypothesis of your study was not amply justified in the introduction.

Why this particular period? When starting the discussion, the authors resume two studies that analyzed the period of time worked on the manuscript under other conditions. Perhaps this explanation makes more sense in the introduction, to justify choosing this time period for analyzing your data.

In the method, I believe that the authors could try to explain more clearly the composition of the time points worked, as well as better explain if the same people were evaluated at the two time points. Remember that the reader needs to fully understand what is being worked on.

In the third line of the first paragraph of the methods, I believe there is no need to repeat the information: "diagnostic criteria for ASD require symptoms to have developed within one month of trauma)", since the important thing is that this is well detailed in the introduction.

Regarding the second paragraph of the methods, I felt a lack of additional information, such as: what is the recruitment period? How did the data collection take place? who carried out the recruitment? where was the study carried out, where were the tests applied?

Still in the second paragraph (2nd line), the authors report: "Stroke Unit of a tertiary hospital which serves a mixed urban and rural catchment area of approximately 1 million people". In what period of time is the service to this public?

In addition, the authors report: "Exclusion criteria were (1) too medically unwell as determined by a stroke physician" (line 5). What would be considered "too medically unwell"? It is important that this information is clear, so that any researcher in the world can reproduce the research design.

In the third paragraph, the authors start saying: "From study inception date". But it is unclear what the start date was. I believe it is important to provide this information. In addition, I consider it important that the authors report how they arrived at the sample N? Was it for convenience? Was a sample calculation performed? Were possible sample losses considered? There was a lot of sample loss, what strategies were used to minimize friction and increase participant adherence?

I am concerned about the analysis q was performed for such a small n, considering the large number of independent variables that were analyzed. Have tests to assess the fit of the data to the model been performed?

In the fifth paragraph of the method, the text refers to the final N of 54 participants. However, at the second time point this N decreased. The same is portrayed in the constructed tables (reporting an N of 54 participants). However, if the analysis was carried out at two time points, and at the second time point there were not 54 participants for the final analysis, I believe that the right thing would be to put the information and perform the analyzes on the final participants, who actually participated in the interventions. . I suggest that the authors re-evaluate this and make this information clearer in the text.

In the procedure subtopic, I felt a lack of information, such as: how long did the tests take? where were the tests performed? how were participants contacted at different time points?

In the second paragraph of the subtopic "Procedures" the authors suggest that the analysis performed was a linear regression. Which is not in accordance with what is presented in the abstract and in the tables presented. I suggest that this be reviewed.

In the subtopic "Hypothesis Testing" the authors mention what was worked on in blocks 1, 2 and 3. There is a description of the variables analyzed in block 1, but not in blocks 2 and 3, the authors directly report the results found. I suggest that this be readjusted, in order to make the relationship between the variables worked in each block clear.

At some points in the discussion, I believe that the writing is confused about when the authors were talking about the results of the study or other studies in the literature. I believe it is necessary to re-evaluate this throughout the topic.

Regarding the references, about 81% of them are outdated. It is suggested that at least 75% of the references used are current, especially in the introduction, to reflect the current panorama of the phenomenon being analyzed. Thus, I suggest that authors can update the references.

Reviewer #2: While in general I find it good that the authors have examined risk factors for PTSD development after stroke I feel that there a number of problems with the ms that need to be addressed before it can be decided whether it should be published or not.

The attrition rate is substantial in this study. A large number of exclusion criteria takes us to a starting sample of mild to severe cases who are able to communicate in English - only a fourth of the starting population. I believe these limitations should be pointed out more clearly. Then, 54 down to 31 in the final step (57% participation in the follow-up) is a rather big drop-out, and a substantial part of those who abstained from the follow-up may have done so (refusals and no responses) because of the large number of questionnaires they were exposed to.

We need a column in table 1 showing the basic characteristics of those who participated in the follow-up so we can compare follow-up participants with the total group.

It is not clear to me how NARS can be used reliably in a sample of stroke patients. And the authors do not use it for any prediction either. It can be omitted from the ms. On the other hand it would have been interesting to know more about the patients´social situation. The authors show education level but that does not tell an international readership so much. What is the average education level in a sample with this age and gender composition in Ireland? Another set of potentially really important predictors would have been social variables, such as employment status.

In my understanding it is really hard to generalise anything from this study

Reviewer #3: The abstract does not point out important information about the method. The loss of participants between the beginning and the end of the experiment was very high, even compromising the statistical analysis. The discussion was not considered sufficient to corroborate the data obtained in the research.

6. PLOS authors have the option to publish the peer review history of their article (what does this mean?). If published, this will include your full peer review and any attached files.

Reviewer #1: No

Reviewer #2: No

Reviewer #3: No

---

## [Author Response · Author response to Decision Letter 0]

14 Apr 2023

The authors are sincerely grateful to the reviewers for their careful and very helpful suggestions for ways to improve the manuscript. Please see our replies below and revisions within the manuscript, all in red font for ease of review.

Reviewer #1: 

1. In the intro part, some points need to be readjusted. I believe that there was an error in the presentation of the references throughout the text, since they do not follow a citation order, starting in the text with reference four, as presented in the first line of the first paragraph of the introduction.

Reply: Apologies for this error – the referencing has been corrected.

2. Between the second and third paragraph I missed a link about stroke. Why study this condition? The inclusion of epidemiological data could help in this justification.

Reply: We have reorganised the Introduction and added information to justify the relevance of the study as follows: 

ASD is differentiated from PTSD in the psychiatric classification system primarily by virtue of the duration of the symptoms. The symptoms of ASD must persist for at least 3 days and must resolve within one month of the trauma – if the symptoms persist for more than one month, then PTSD may be diagnosed. It is possible that PTSD may develop at any stage after a traumatic event, although normally symptoms appear within a few months of the event [1]. ASD is a risk factor for PTSD but it is not necessary for traumatic stress (ASD) symptoms to have been present immediately after the event in order for PTSD to develop subsequently [2]. 

Recent reviews highlighted that relatively little is still known about the development of specific types of anxiety [3] and about traumatic stress reactions, other than PTSD, following stroke [4]. Previous studies suggest that as many as one in four people may develop PTSD after a stroke [5] with factors such as younger age [6] and co-morbid depression and anxiety along with negative appraisals and perceived high personal risk of another stroke [7] conferring higher risk of PTSD post-stroke. As psychological difficulties can impact on functional outcomes [8] and quality of life in stroke survivors [9], a call has been made for longitudinal studies to aid in the early identification of risk factors for PTSD in the aftermath of stroke [10] – one such risk factor could include the presence of ASD symptoms [11]. A recent review [12] has examined methods for preventing the development of PTSD and identified psychological treatments such as trauma-focused cognitive therapy (CBT-T) that can both ameliorate the effects of ASD and reduce the risk of PTSD if offered in the first six months after a traumatic event, but none of the studies were specific to a stroke population. 

Thus far, there is an absence of research examining (a) the extent to which ASD manifests after a stroke and (b) the potential influence of ASD symptoms as a risk factor for developing PTSD in stroke survivors. Based on evidence-based practice guidelines recommending early treatment of ASD [13,14] it seems plausible that psychological interventions to identify and treat acute stress symptoms after stroke may reduce the potential for these symptoms to develop into a more chronic psychological condition. 

3. It is important to create a logical line of reasoning during the text. In the third paragraph it starts a new content and in the fourth paragraph it goes back to the content worked in the second. This makes the text confusing. I suggest that authors can readjust, considering the construction of the introduction following the funnel model, starting from the most general subject to the most specific.

Reply: We have made modifications to the text and hope that the linkage and flow of ideas is improved.

4. I felt that the hypothesis of your study was not amply justified in the introduction.

Why this particular period? When starting the discussion, the authors resume two studies that analyzed the period of time worked on the manuscript under other conditions. Perhaps this explanation makes more sense in the introduction, to justify choosing this time period for analyzing your data.

Reply: We have clarified the basis for our hypotheses as follows on page 3: 

ASD is a known risk factor for PTSD [4,13,14] and their respective diagnosis is based on duration of symptoms following exposure to the traumatic event. Thus, it was hypothesized that higher levels of ASD symptoms at 1-2 weeks post stroke would predict higher levels of PTSD symptoms at 6-12 weeks post-stroke.

5. In the method, I believe that the authors could try to explain more clearly the composition of the time points worked, as well as better explain if the same people were evaluated at the two time points. Remember that the reader needs to fully understand what is being worked on.

Reply: The same patients were followed across the two time points – we hope this is now clear at various places in the manuscript and in the revised Method (page 4):

A prospective cohort study was carried out to investigate whether acute stress symptoms measured at baseline predicted post-traumatic stress symptoms at follow-up. Suitable patients were assessed at Time 1 (baseline) within two weeks of having a stroke (diagnostic criteria for ASD require symptoms to have developed within one month of trauma). The same patients were assessed again at Time 2 between 6-12 weeks after their stroke (diagnostic criteria for PTSD require that symptoms have persisted for more than one month).

6. In the third line of the first paragraph of the methods, I believe there is no need to repeat the information: "diagnostic criteria for ASD require symptoms to have developed within one month of trauma)", since the important thing is that this is well detailed in the introduction.

Reply: Since this distinction is in essence what differentiates ASD and PTSD, we have left it in the Method just for clarity for the reader.

7. Regarding the second paragraph of the methods, I felt a lack of additional information, such as: what is the recruitment period? How did the data collection take place? who carried out the recruitment? where was the study carried out, where were the tests applied?

Still in the second paragraph (2nd line), the authors report: "Stroke Unit of a tertiary hospital which serves a mixed urban and rural catchment area of approximately 1 million people". In what period of time is the service to this public?

We have added more information (copied below) and hope we have addressed the reviewer’s questions: 

Patients were enrolled over a 12-month period (January 2019 – January 2020) at a regional Stroke Unit in the West of Ireland, which serves a mixed urban and rural catchment area of approximately 1 million people. Potentially suitable patients were identified by the physicians and nurses on the Stroke Team. The participants were essentially a “convenience sample” in that they were the patients admitted to hospital (as opposed to those seen in ER and discharged without admission) and those attending this particular hospital – there are several other hospitals in the region although the hospital where the current study was undertaken is the only one with a dedicated Stroke Unit. The duration of recruitment to the study was based on logistical factors such as the contract duration of research personnel, rather than on sample size estimations.

8. In addition, the authors report: "Exclusion criteria were (1) too medically unwell as determined by a stroke physician" (line 5). What would be considered "too medically unwell"? It is important that this information is clear, so that any researcher in the world can reproduce the research design.

Reply: We mean a patient who is effectively unconscious or so impaired that they could not engage with the researcher. It is difficult to be more specific because each patient may present with different obstacles such as inability to speak, too cognitively impaired, requiring mechanical ventilation etc. We hope this clarifies what we mean. We have revised the text as follows: (1) too medically unwell as determined by a stroke physician (e.g. reduced consciousness or otherwise unable to interact with researcher) 

9. In the third paragraph, the authors start saying: "From study inception date". But it is unclear what the start date was. I believe it is important to provide this information. In addition, I consider it important that the authors report how they arrived at the sample N? Was it for convenience? Was a sample calculation performed? Were possible sample losses considered? There was a lot of sample loss, what strategies were used to minimize friction and increase participant adherence?

Reply: Thank you for raising these important questions. We have provided additional data on page 4 about the dates of the study and the sampling constraints, as follows:

Patients were enrolled over a 12-month period (January 2019 – January 2020) at a regional Stroke Unit in the West of Ireland, which serves a mixed urban and rural catchment area of approximately 1 million people. Potentially suitable patients were identified by the physicians and nurses on the Stroke Team. The participants were essentially a “convenience sample” in that they were the patients admitted (as opposed to those seen in ER and discharged without admission) and those attending this particular hospital – there are several other hospitals in the region although the hospital where the current study was undertaken is the only one with a dedicated Stroke Unit. The duration of recruitment to the study was based on temporal factors such as the contract duration of research personnel, rather than on sample size estimations. 

These weaknesses have also been highlighted in the Discussion.

10. I am concerned about the analysis q was performed for such a small n, considering the large number of independent variables that were analyzed. Have tests to assess the fit of the data to the model been performed?

Reply: We assessed the fit of the data to the model using R2. We have augmented this now by also considering and reporting on Adjusted R2, with the latter increasing significantly from Block 2 to Block 3. We also added more information from the ANOVA table 4 to improve the completeness of reporting. 

11. In the fifth paragraph of the method, the text refers to the final N of 54 participants. However, at the second time point this N decreased. The same is portrayed in the constructed tables (reporting an N of 54 participants). However, if the analysis was carried out at two time points, and at the second time point there were not 54 participants for the final analysis, I believe that the right thing would be to put the information and perform the analyzes on the final participants, who actually participated in the interventions. I suggest that the authors re-evaluate this and make this information clearer in the text.

Reply: Thank you for pointing this out – we acknowledge it could cause confusion. We would like to confirm that the Time 2 analyses were performed on those available at Time 2 (n=31). We have revised the text – highlighted in relevant parts of the Method and Results – to improve clarity. We have also expanded Table 1 to provide the data for the n=31 respondents at Time 2.

12. In the procedure subtopic, I felt a lack of information, such as: how long did the tests take? where were the tests performed? how were participants contacted at different time points?

Reply: We have added more information on page 5: 

At Time 1 the assessments were carried out in the Stroke Unit setting. At Time 2, the assessments were carried out in various locations, depending on where the patient was at that time (variously people were seen in the same hospital, at a rehabilitation hospital, in a nursing home, at an outpatients clinic or in their own home). The measures that were administered are listed below and took an average of 45-60 minutes to complete, testing was sometimes carried out over two or more occasions as determined by the patient’s capacity to stay engaged.

13. In the second paragraph of the subtopic "Procedures" the authors suggest that the analysis performed was a linear regression. Which is not in accordance with what is presented in the abstract and in the tables presented. I suggest that this be reviewed.

Reply: Thank you. We have now reviewed and amended the text. 

14. In the subtopic "Hypothesis Testing" the authors mention what was worked on in blocks 1, 2 and 3. There is a description of the variables analyzed in block 1, but not in blocks 2 and 3, the authors directly report the results found. I suggest that this be readjusted, in order to make the relationship between the variables worked in each block clear.

Reply: The Hypothesis Testing section has been amended and we hope the procedure is now more clear.

15. At some points in the discussion, I believe that the writing is confused about when the authors were talking about the results of the study or other studies in the literature. I believe it is necessary to re-evaluate this throughout the topic.

Reply: Thank you for pointing this out. We have reviewed and expanded the Discussion and have endeavoured to be clear about reference to our own study or to other studies.

16. Regarding the references, about 81% of them are outdated. It is suggested that at least 75% of the references used are current, especially in the introduction, to reflect the current panorama of the phenomenon being analyzed. Thus, I suggest that authors can update the references.

Reply: We have searched for updated literature and included 10 new references – it is not always possible to find newer references if work has not been carried out in the area or if older work is (a) seminal or (b) more relevant. New references are highlighted in the reference list.

Reviewer #2: 1. While in general I find it good that the authors have examined risk factors for PTSD development after stroke I feel that there a number of problems with the ms that need to be addressed before it can be decided whether it should be published or not.

The attrition rate is substantial in this study. A large number of exclusion criteria takes us to a starting sample of mild to severe cases who are able to communicate in English - only a fourth of the starting population. I believe these limitations should be pointed out more clearly. Then, 54 down to 31 in the final step (57% participation in the follow-up) is a rather big drop-out, and a substantial part of those who abstained from the follow-up may have done so (refusals and no responses) because of the large number of questionnaires they were exposed to.

Reply: These are all valid points and are weaknesses of the study. We have made these issues more explicit at various points highlighted in the revised paper including the Method and Discussion. Unfortunately the exclusions were reflective of the many problems encountered by the clinical population under study and is a difficulty in carrying out field-based opportunistic research of this nature. Despite these limitations, we believe the study does provide some clinically important insights.

2. We need a column in table 1 showing the basic characteristics of those who participated in the follow-up so we can compare follow-up participants with the total group.

Reply: Thank you for this very helpful suggestion. Table 1 has been revised accordingly. 

3. It is not clear to me how NARS can be used reliably in a sample of stroke patients. And the authors do not use it for any prediction either. It can be omitted from the ms. On the other hand it would have been interesting to know more about the patients´ social situation. The authors show education level but that does not tell an international readership so much. What is the average education level in a sample with this age and gender composition in Ireland? Another set of potentially really important predictors would have been social variables, such as employment status. In my understanding it is really hard to generalise anything from this study.

Reply: Again we appreciate these important comments and suggestions and we have attempted to provide more contextual information in the Discussion to help readers judge the extent to which the findings are generalisable. We also respectfully point out that Table 1 contains information on social factors including marital status and living situation but we did not collect information on employment status. As suggested, we have removed reference to the NART.

Reviewer #3: 

1. The abstract does not point out important information about the method. 

Reply: We are uncertain what important information is missing but hope the revised document in general provides better clarity.

2. The loss of participants between the beginning and the end of the experiment was very high, even compromising the statistical analysis. 

Reply: We acknowledge this weakness and have ensured it is clearly described in the Discussion.

3. The discussion was not considered sufficient to corroborate the data obtained in the research.

Reply: We are uncertain what specifically the reviewer is concerned about but we have carefully reviewed the Discussion to include updated literature and we have revised any text which could be considered as over-reaching.

---

## [Decision Letter · Decision Letter 1]

11 May 2023

Acute stress symptoms 1-2 weeks after stroke predict the subsequent development of post-traumatic stress symptoms: A prospective cohort study.

PONE-D-22-27178R1

Dear Dr. McGuire,

Please remember to provide their data / data availability statement.

We’re pleased to inform you that your manuscript has been judged scientifically suitable for publication and will be formally accepted for publication once it meets all outstanding technical requirements.

Kind regards,

Thiago P. Fernandes, PhD

Academic Editor

PLOS ONE

Additional Editor Comments (optional):

Reviewers' comments:

Reviewer's Responses to Questions

**Comments to the Author**

1. If the authors have adequately addressed your comments raised in a previous round of review and you feel that this manuscript is now acceptable for publication, you may indicate that here to bypass the “Comments to the Author” section, enter your conflict of interest statement in the “Confidential to Editor” section, and submit your "Accept" recommendation.

Reviewer #1: All comments have been addressed

Reviewer #2: All comments have been addressed

2. Is the manuscript technically sound, and do the data support the conclusions?

Reviewer #1: Yes

Reviewer #2: Yes

3. Has the statistical analysis been performed appropriately and rigorously? 

Reviewer #1: Yes

Reviewer #2: Yes

4. Have the authors made all data underlying the findings in their manuscript fully available?

Reviewer #1: Yes

Reviewer #2: No

5. Is the manuscript presented in an intelligible fashion and written in standard English?

Reviewer #1: Yes

Reviewer #2: Yes

6. Review Comments to the Author

Reviewer #1: I would like to congratulate the authors for the care with the considerations provided in the opinion.

I believe that all suggestions were attended to as much as possible and the limitations were well described at the end of the text. Thus, I consider that the article is fit for publication.

Reviewer #2: The authors have addressed the comments well. The weaknesses have been pointed out. Despite those, the main point is well taken. I looked for any statement regarding data availability but did not find any. This has to be supplemented, according to the journal´s policy

7. PLOS authors have the option to publish the peer review history of their article (what does this mean?). If published, this will include your full peer review and any attached files.

Reviewer #1: No

Reviewer #2: **Yes: **Töres Theorell

---

## [Editor Report · Acceptance letter]

5 Jun 2023

PONE-D-22-27178R1 

Acute stress symptoms 1-2 weeks after stroke predict the subsequent development of post-traumatic stress symptoms: A prospective cohort study. 

Dear Dr. McGuire:

I'm pleased to inform you that your manuscript has been deemed suitable for publication in PLOS ONE. Congratulations! Your manuscript is now with our production department. 

Kind regards, 

on behalf of

Dr. Thiago P. Fernandes 

Academic Editor

PLOS ONE